

# Development of an expert system for the classification of myalgic encephalomyelitis/chronic fatigue syndrome

Fatma Hilal Yagin[1], Ahmadreza Shateri[2], Hamid Nasiri[3], Burak Yagin[1], Cemil Colak[1] and Abdullah F. Alghannam[4]

[1] Department of Biostatistics and Medical Informatics, Inonu University, Malatya, Türkiye
[2] Electrical and Computer Engineering Department, Semnan University, Semnan, Iran
[3] Department of Computer Engineering, Amirkabir University of Technology (Tehran Polytechnic), Tehran, Iran
[4] Lifestyle and Health Research Center, Princess Nourah bint Abdulrahman University, Riyadh, Saudi Arabia

## ABSTRACT

Myalgic encephalomyelitis/chronic fatigue syndrome (ME/CFS) is a severe condition with an uncertain origin and a dismal prognosis. There is presently no precise diagnostic test for ME/CFS, and the diagnosis is determined primarily by the presence of certain symptoms. The current study presents an explainable artificial intelligence (XAI) integrated machine learning (ML) framework that identifies and classifies potential metabolic biomarkers of ME/CFS. Metabolomic data from blood samples from 19 controls and 32 ME/CFS patients, all female, who were between age and body mass index (BMI) frequency-matched groups, were used to develop the XAI-based model. The dataset contained 832 metabolites, and after feature selection, the model was developed using only 50 metabolites, meaning less medical knowledge is required, thus reducing diagnostic costs and improving prognostic time. The computational method was developed using six different ML algorithms before and after feature selection. The final classification model was explained using the XAI approach, SHAP. The best-performing classification model (XGBoost) achieved an area under the receiver operating characteristic curve (AUCROC) value of 98.85%. SHAP results showed that decreased levels of alpha-CEHC sulfate, hypoxanthine, and phenylacetylglutamine, as well as increased levels of N-delta-acetylornithine and oleoyl-linoloyl-glycerol (18:1/18:2)[2], increased the risk of ME/CFS. Besides the robustness of the methodology used, the results showed that the combination of ML and XAI could explain the biomarker prediction of ME/CFS and provided a first step toward establishing prognostic models for ME/CFS.

## INTRODUCTION

Myalgic encephalomyelitis/chronic fatigue syndrome (ME/CFS) is a debilitating disease of unknown cause and poor prognosis. There is currently no specific diagnostic test for

Corresponding authors
Fatma Hilal Yagin,
hilal.yagin@inonu.edu.tr
Burak Yagin,
burak.yagin@inonu.edu.tr

ME/CFS, and the diagnosis is made based on the presence of characteristic symptoms, such as severe fatigue, post-exertional malaise, cognitive impairment, and unrefreshing sleep, that have persisted for at least six months (*Deumer et al., 2021*).

As a growing body of research has identified abnormalities in the gut microbiome, immune system, neuroimaging, exercise physiology, and blood metabolites of ME/CFS patients. However, the underlying cause of the disease remains unclear, and there is no consensus among researchers regarding the primary pathophysiological mechanisms involved (*Missailidis, Annesley & Fisher, 2019*). Some hypotheses suggest that ME/CFS may be caused by an infectious agent, such as a virus (*Rasa et al., 2018*) or bacteria, or by an autoimmune response to an infection or environmental trigger. Others propose that the disease may be related to abnormalities in mitochondrial function, impaired cellular metabolism, or dysregulation of the autonomic nervous system (*Komaroff & Lipkin, 2021*). Despite these various theories, there is still much to be learned about the underlying mechanisms of ME/CFS, and much more research is needed to develop effective diagnostic and treatment strategies for this complex and debilitating disease.

Metabolomics technology has the potential to improve the diagnosis and management of ME/CFS, as well as to enhance understanding of the underlying pathophysiology of the disease. By identifying specific metabolites or metabolic pathways that are deregulated in ME/CFS, may be able to develop more targeted treatments and interventions that address the underlying biochemical imbalances. However, more research is needed to validate the use of metabolomics biomarkers for ME/CFS and to identify specific metabolites or pathways that are consistently deregulated across different patient populations and disease stages (*Che et al., 2022*; *Germain et al., 2022*).

*Germain et al. (2018)* conducted a metabolomics investigation to discover potential biomarkers for ME/CFS. They used plasma samples from both ME/CFS patients and healthy individuals and utilized mass spectrometry to analyze the metabolite levels. The outcomes revealed that there were significant discrepancies in the levels of metabolites linked to oxidative stress and antioxidant defenses between the ME/CFS patients and the healthy controls. The research also recognized some promising biomarkers for ME/CFS, including metabolites associated with the pentose phosphate pathway, oxidative stress response, and glutathione metabolism. The results imply that a redox imbalance could be a critical element in the pathophysiology of ME/CFS. Overall, the study provides supportive data for potential biomarkers and fundamental mechanisms of ME/CFS, which could contribute to better diagnosis and treatment of this multifaceted and incapacitating illness (*Germain et al., 2018*).

In this study, a machine learning (ML) framework combined with explainable artificial intelligence (XAI) is proposed to extract potential metabolite biomarkers for ME/CFS diagnosis and monitoring. The model preprocesses the data and then applies ANOVA based on the F-value of the metabolite features to extract the significantly expressed metabolites between the two classes (ME/CFS) and healthy control samples. The biomarkers metabolite feed the Extreme Gradient Boosting (XGBoost) classifier to predict the class of the samples. For most of the performance measurements, the proposed model outperformed the

standard classifiers, including random forest, decision tree, support vector classifier (SVMs), Naïve Bayes, and logistic regression.

## MATERIAL AND METHODS

### Study design, data, and compliance with ethical standards

In the open-access metabolomics dataset used in this study, the cohort consisted of 19 controls and 32 patients, all female, who were between age and body mass index (BMI) frequency-matched groups. The blood samples were collected in Ethylene Diamine Tetra Acetic Acid (EDTA) tubes. Then, plasma was separated from the cells through centrifugation at 500 g for 30 min and stored at a temperature of $-80\,°C$ for further analysis. After that, global metabolomics was performed using four ultra-high-performance liquid chromatography/tandem accurate mass spectrometry (UHPLC/MS/MS). This automated service allows for the precise measurement of hundreds of metabolites in a wide range of categories (*Germain et al., 2018*). A total of 42,432 data points from 51 participants were collected using the Metabolon® technology to identify and analyze 832 different metabolites. Amino acids (177), carbohydrates (26), cofactors and vitamins (28), energy (10), lipids (353), nucleotides (29), peptides (33), and xenobiotics (176) are among the kinds of metabolites that this study examines. Information about the metabolomics data set, which is available as Supplemental File 1, contains information about the 83 sub-pathways that can be further separated into the eight super-pathways that Metabolon® has identified.

The sample size required for this study was estimated with MetSizeR based on the PPCA model and calculated by setting the false discovery rate to 0.05. As a result, a minimum sample size of 28 patients in total with 14 patients in each group was estimated. Although it was challenging to find ME/CFS patients and healthy controls who satisfied the study's inclusion requirements, the sample size was more than that predicted by MetSizeR (*Nyamundanda et al., 2013*), a technique used to estimate sample size in metabolomics investigations. The Institutional Review Board for Non-Interventional Clinical Research at Inonu University gave ethical permission to this study (decision no. 2023/4512).

### Methods

The schematic representation of the methodology used and proposed in the research is as in Fig. 1.

**Feature selection**: Feature selection helps reduce the dimensionality of a dataset, making it easier to interpret and faster to process. It can also improve the performance of machine learning models by removing irrelevant or redundant features that can lead to overfitting (*Chandrashekar & Sahin, 2014*; *Guyon & Elisseeff, 2003*).

Analysis of Variance (ANOVA) feature selection is a statistical method used to identify the most important features (or variables) that are most strongly associated with the target variable. The method involves calculating the F-value for each feature, which measures how much the variation in the target variable can be explained by the variation in the feature. The higher the F-value, the more significant the feature is thought to be in predicting the target variable (*Nasiri & Alavi, 2022*).

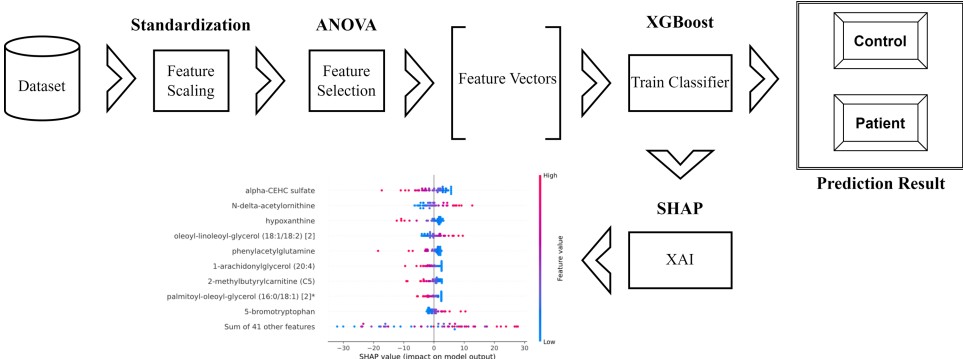

**Figure 1** Schematic representation of the proposed method.

The F-value is calculated as the ratio of the mean square between groups (MSB) to the mean square within groups (MSW):

$$F = \frac{MS_{between}}{MS_{within}} \tag{1}$$

where:

$$MSB = \frac{\sum(\overline{x}_k - \overline{x})^2 n_k}{df_{between}} \tag{2}$$

$n_k$ denotes the number of measurements in the $k$th class, $\overline{x}_k$ denotes the mean of samples for each class, and $\overline{x}$ denotes the mean of entire samples.

The mean square within groups is calculated as:

$$MSW = \frac{\left(\sum\sum(x_{nk} - \overline{x}_k)^2\right)}{df_{within}} \tag{3}$$

$x_{nk}$ represents the $n$th value for the $k$th class.

The degrees of freedom are calculated as:

$$df_{between} = K - 1 \tag{4}$$

$$df_{within} = N - K \tag{5}$$

$K$ and $N$ are symbols for the number of classes and the overall sample size, respectively (*Johnson & Synovec, 2002*).

**Extreme gradient boosting (XGBoost)**: XGBoost is a popular machine learning algorithm that belongs to the gradient boosting family of algorithms. It was developed by *Chen & Guestrin (2016)* in 2016 and has since become widely used in the machine-learning community due to its efficiency, scalability, and high level of performance on structured data problems (*Cao et al., 2022*; *Ghaheri et al., 2023*; *Homafar, Nasiri & Chelgani, 2022*).

The basic equation for XGBoost can be written as follows:

$$\hat{y}_i = \sum_{k=1}^{K} f_k(x_i) \tag{6}$$

where $\hat{y}_i$ isthe predicted value for the $i$th instance, $K$ is the number of weak models, $f_k(x_i)$ isthe output of the $k$th weak model on the $i$th instance, and $x_i$ isthe feature vector for the $i$th instance (*Farzipour, Elmi & Nasiri, 2023*; *Nasiri, Homafar & Chelgani, 2021*).

The weak models in XGBoost are decision trees, where each tree predicts the residual error of the previous tree. This approach is called gradient boosting, as the subsequent models try to minimize the gradient of the loss function concerning the predicted values (*Ayyadevara & Ayyadevara, 2018*; *Chelgani et al., 2023*). The learning objective function for XGBoost is a regularized version of the loss function, which helps prevent overfitting and improves generalization (*Liu et al., 2021*; *Maleki, Raahemi & Nasiri, 2023*).

The objective function that XGBoost minimizes can be written as:

$$Obj(\theta) = \sum_{i=1}^{n} l(\hat{y}_i, y_i) + \sum_{k=1}^{K} \Omega(f_k) \tag{7}$$

where $\theta$ denotes the model parameters, $n$ is the number of instances, $l$ is the loss function that measures the difference between the predicted and true values, and $\Omega$ is the penalizing regularization function for complicated models (*Maleki, Raahemi & Nasiri, 2023*) and is computed as:

$$\Omega(f_k) = \gamma T + \frac{1}{2}\lambda\|\omega\|^2 \tag{8}$$

where $\gamma$ and $\lambda$ are variables that control the penalty related to the quantity of leaves T and the weight of each leaf $\omega$, respectively (*Nasiri & Hasani, 2022*; *Nasiri, Homafar & Chelgani, 2021*).

**Random forest**: Random forest is an ensemble learning algorithm first proposed by Leo Breiman in 2001 (*Breiman, 2001*). The algorithm is based on the idea of combining multiple decision trees to create a more robust and accurate model (*Rodriguez-Galiano et al., 2012*).

Each decision tree is trained on a random subset of the training data and a random subset of the features at each split (*Gong et al., 2018*). This introduces randomness into the training process, which helps reduce overfitting and improve the model's generalization performance (*Alam & Vuong, 2013*). At prediction time, each tree in the forest independently predicts a class or a numerical value, and the final prediction is obtained by combining the individual predictions, for example, by taking the majority vote (classification) or the mean value (regression) (*Breiman, 2001*).

One of the critical advantages of random forest is its ability to estimate feature importance, which can be useful for understanding the underlying patterns in the data and selecting the most relevant features for the task (*Li, Harner & Adjeroh, 2011*). The feature importance is calculated by measuring the decrease in performance (*e.g.*, accuracy or mean squared error) when a particular feature is randomly permuted, which provides an estimate of how much the feature contributes to the model's predictive power.

**Support vector classifier**: A support vector classifier (SVMs) is a machine learning algorithm that is used for classification tasks. The SVMs are based on SVMs theory, which *Vapnik (1999)* introduced in the 1990s.

The idea behind the SVMs is to find the best possible hyperplane that separates two classes of data points in a high-dimensional space (*Khairnar & Kinikar, 2013*). The hyperplane is chosen to maximize the distance between the hyperplane and the closest data points from each class (known as support vectors) (*Pradhan, 2012*). This distance is known as the margin, and the SVM/SVMs is often referred to as a maximum-margin classifier (*Amarappa & Sathyanarayana, 2014*).

To train SVMs with a linear kernel, the algorithm first identifies the support vectors, which are the data points closest to the decision boundary (*Alam et al., 2020*). Then, the algorithm finds the optimal hyperplane that maximizes the distance between the support vectors of each class (*Luta, Baldovino & Bugtai, 2018*). During the testing phase, new data points are classified based on which side of the hyperplane they fall on (*Amarappa & Sathyanarayana, 2014*). One of the advantages of using SVMs with a linear kernel is that it can work well even in high-dimensional spaces, where the number of features is much greater than the number of observations. However, it may not perform well when the data is not linearly separable. In such cases, a non-linear kernel can be used instead (*Ghosh, Dasgupta & Swetapadma, 2019*).

**SHapley Additive exPlainations (SHAP)**: SHAP is a method for explaining the output of any machine learning model. It was introduced by *Lundberg et al. (2018)* and it provides a way to calculate the contribution of each feature to the prediction made by the model (*Štrumbelj & Kononenko, 2014*).

The SHAP method is based on the concept of Shapley values, which is a well-known concept in the cooperative game theory (*Ekanayake, Meddage & Rathnayake, 2022*). The idea is to calculate the marginal contribution of each feature to the prediction by considering all possible subsets of features that could have been included in the model (*Fatahi et al., 2023*; *Li, 2022*).

The main equation for calculating SHAP values is:

$$\phi_i(x) = \sum_{S \subseteq N \setminus \{i\}} \frac{|S|!(|N| - |S| - 1)!}{|N|!} [f_{S \cup \{i\}} - f_S] \tag{9}$$

where:

$\phi_i(x)$ is the SHAP value for feature $i$ of instance $x$

$N$ is the set of all features

$S$ is a subset of features, excluding $i$

$|S|$ is the number of features in subset $S$

$|N|$ is the total number of features

$f_{S \cup \{i\}}$ is the output of the model when features in $S$ are present along with feature $i$

$f_S$ is the output of the model when only features in $S$ are present (*Lundberg & Lee, 2017*).

This equation calculates the contribution of feature $i$ by considering all possible subsets of features that could have been included in the model. It calculates the difference between the model's output when feature $i$ is present and the output when feature $i$ is absent, averaging over all possible combinations of features that include feature $i$. This results in a measure of the marginal contribution of feature $i$ to the prediction (*Bi et al., 2020*).

In practice, SHAP values can be calculated using various methods, including tree-based algorithms and kernel-based methods. These values can then be used to explain the output of a model and to identify which features are most important for making predictions (*Mangalathu, Hwang & Jeon, 2020*).

**Performance assessment:** In the presented study, individuals are classified as either having or not having ME/CFS based on the extracted metabolic biomarkers. Predictions can therefore be divided into four groups: True Positive (TP) signifies correctly identifying individuals with ME/CFS, True Negative (TN) denotes the accurate identification of individuals without ME/CFS, False Positive (FP) involves misclassifying individuals without ME/CFS as having the condition, and False Negative (FN) involves misclassifying individuals with ME/CFS as not having the condition. The following metrics are used to evaluate the classification results:

$$Precision = \frac{TP}{TP + FP} \tag{10}$$

$$Sensitivity = \frac{TP}{TP + FN} \tag{11}$$

$$Specificity = \frac{TN}{TN + FP} \tag{12}$$

$$F1 - score = \frac{2}{\frac{1}{Sensitivity} + \frac{1}{precision}} \tag{13}$$

$$Accuracy = \frac{TP + TN}{TP + FN + FP + TN} \tag{14}$$

Another performance metric utilized was the Brier score, which measures the accuracy of the model's probability predictions.

$$BS = \frac{1}{N} \sum_{t=1}^{N} (f_t - o_t)^2 \tag{15}$$

In which $f_t$ denotes the probability that was forecast, $o_t$ represents the actual outcome of the event, and $N$, and N is the number of forecasting instances.

# RESULTS

In the current study, the dataset, derived from metabolomic analyses, comprised 51 cases, of which 19 were healthy controls and the remaining 32 were patients diagnosed with ME/CFS. Initially, a standardization process was employed to scale the features. This step was crucial to ensure equitable contribution of each feature to the analysis and to mitigate the undue influence of any single feature, particularly in the presence of outliers. Subsequently, the dataset, characterized by an extensive set of 832 features, necessitated the application of ANOVA feature selection. This methodological choice was driven by the dual objectives of preventing model overfitting and reducing the computational burden during training. The ANOVA F-value, employed in this context, serves as an indicator of the correlation between individual features and the target variable. It quantifies this

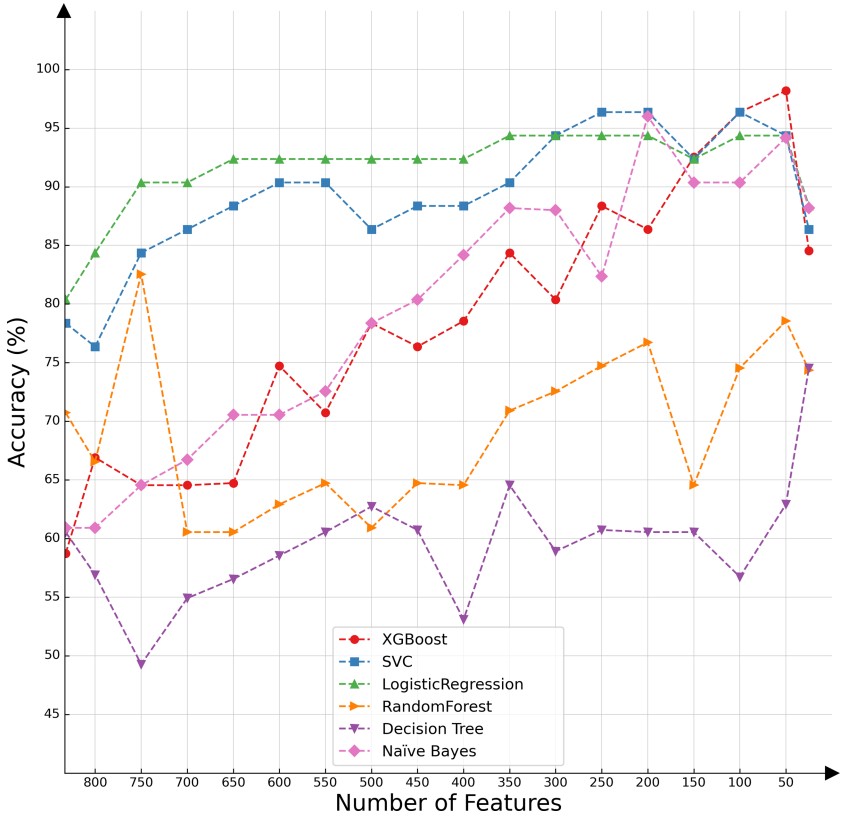

**Figure 2** **Classifier performance based on the number of features.**

relationship by comparing the variance ratio across different feature values (inter-group variance) against the variance within each feature group (intra-group variance). A higher F-value is indicative of a strong correlation between the feature and the target variable, thus signifying its potential significance in the predictive model.

In the present investigation, we conducted a comparative analysis of various classifiers' capabilities in the context of decremental feature groups. The classifiers selected for this assessment included SVMs, logistic regression, XGBoost, random forest, decision tree, and naïve Bayes. The results of the comparative evaluation are succinctly presented in Fig. 2, offering a visual representation of the differential classification efficacies of these algorithms under the specified conditions.

Upon conducting an evaluative analysis of the model's efficacy across varying feature counts, it was discerned that the application of ANOVA, restricted to the top 50 features, optimized the performance of the XGBoost classifier. This optimization is evidenced in Fig. 3, which delineates the metabolites demonstrating the most significant correlation with the ME/CFS status. This figure serves as an illustrative guide, pinpointing those metabolites that hold the greatest relevance in the context of ME/CFS, thereby underlining their potential importance in the diagnostic framework.

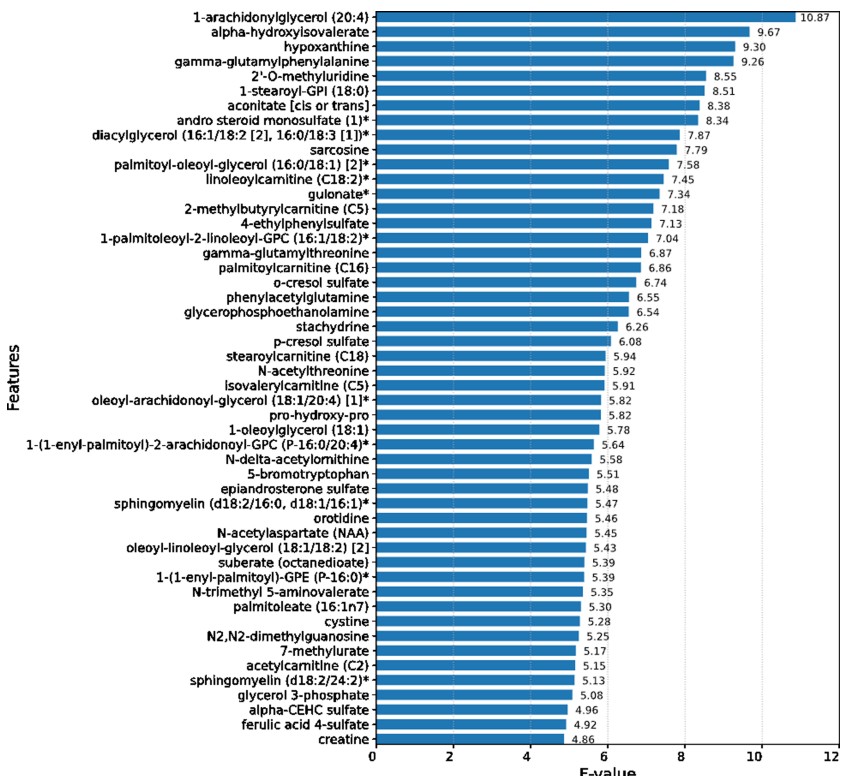

**Figure 3** **Feature importance graph.** The top 50 important metabolites are listed and ordered by ANOVA.

In the study, strategic decision was made to set the booster parameter of the XGBoost algorithm to 'gblinear' during the hyperparameter tuning process. This adjustment aligns with the demonstrated efficacy of linear models in this specific context. Details regarding the modifications made to the XGBoost hyperparameters are comprehensively listed in Table 1, providing a transparent overview of the parameter optimization strategy employed in this study. In the conducted analysis, where models were trained on a dataset comprising 50 features, it was observed that the XGBoost algorithm exhibited superior performance, achieving an accuracy rate of 98.18%. This finding is substantiated in Table 2, which reveals that simpler, more linear models, including logistic regression and SVMs, outperformed their more complex counterparts.

Due to the small dataset, a cross-validation (CV) scheme was used to demonstrate the model's utility and generate precise results that were not overfitted. In 5-fold CV, the data is divided into five equal-sized folds. The model is trained on four folds and tested on the remaining fold. This process is repeated five times, with each fold being used once as the test set. The performance of the model is then averaged over all five folds. The obtained confusion matrices after applying CV are shown in Fig. 4, an overlapped confusion matrix

**Table 1  The XGBoost hyperparameter settings.**

| Hyper-parameter | Value |
| --- | --- |
| Base learner | Gradient boosted linear |
| Tree construction learner | Exact greedy |
| Learning rate ($\eta$) | 0.19 |
| Lagrange multiplier ($\gamma$) | 0 |
| Maximum depth of trees | 6 |

**Table 2  A detailed comparison of before and after feature selection across different evaluation metrics between the proposed method and other classifiers.** Values separated by "/" indicate "without/with" feature selection.

| Model | Sensitivity | Specificity | Precision | F1-Score | Accuracy (%) | Brier score | AUC score |
| --- | --- | --- | --- | --- | --- | --- | --- |
| XGBoost | 0.28 / 1 | 0.84 / 0.95 | 0.80 / 0.97 | 0.41 / 0.98 | 53.27 / 98.18 | 0.479 / 0.023 | 0.74 / 0.99 |
| Random Forest | 0.74 / 0.81 | 0.61 / 0.75 | 0.81 / 0.86 | 0.76 / 0.82 | 70.72 / 78.54 | 0.197 / 0.146 | 0.73 / 0.86 |
| Decision Tree | 0.71 / 0.79 | 0.41 / 0.36 | 0.66 / 0.68 | 0.68 / 0.72 | 60.54 / 62.90 | 0.392 / 0.372 | 0.57 / 0.57 |
| Support Vector Machine | 0.81 / 1 | 0.73 / 0.83 | 0.84 / 0.92 | 0.81 / 0.96 | 78.36 / 94.36 | 0.155 / 0.053 | 0.84 / 0.97 |
| Naïve Bayes | 0.77 / 1 | 0.31 / 0.83 | 0.64 / 0.92 | 0.68 / 0.96 | 60.90 / 94.36 | 0.392 / 0.056 | 0.52 / 0.98 |
| Logistic Regression | 0.88 / 1 | 0.68 / 0.83 | 0.82 / 0.92 | 0.84 / 0.96 | 80.36 / 94.36 | 0.143 / 0.047 | 0.82 / 0.98 |

**Notes.**
Values separated by "/" indicate "without/with" feature selection.

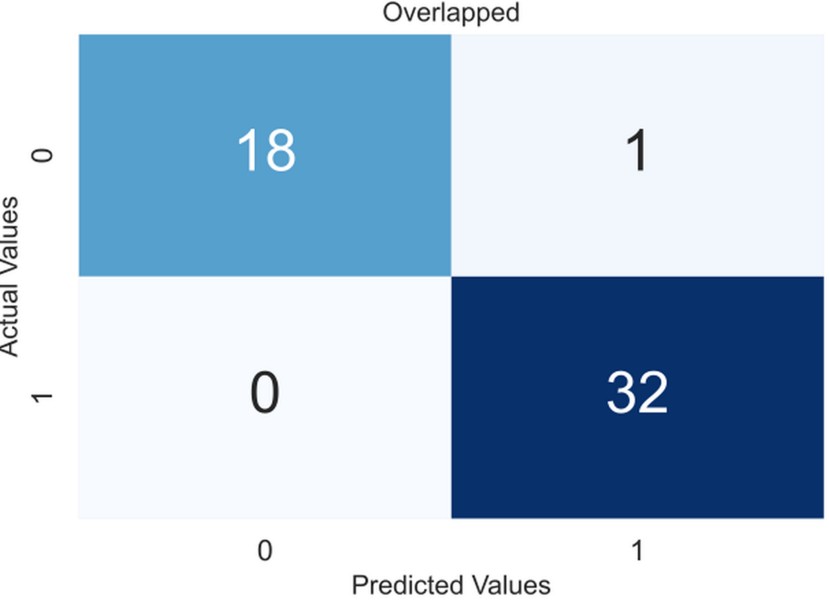

**Figure 4  Confusion matrix for the proposed model (XGBoost).**

to display all classifications. As can be seen, the proposed model had only one type I error.

In order to enhance the robustness and reliability of our model comparisons, we extended our evaluation by performing CV 50 times. The mean, standard deviation, and best results obtained for each model 50 CV times are summarized in Table 3. This

**Table 3 Comparative performance analysis through 50 times cross-validations.** Bold values show the best performance.

| Performance metrics | Model | Mean | Standard deviation | Best result |
|---|---|---|---|---|
| F$_1$-Score (%) | XGBoost | **95.70** | 2.1 | **98.82** |
| | Random Forest | 82.40 | 4.2 | 91.31 |
| | Decision Tree | 69.70 | 5.5 | 83.63 |
| | Support Vector | 93.52 | 2.4 | 97.78 |
| | Naïve Bayes | 94.46 | 1.9 | 98.18 |
| | Logistic Regression | 94.75 | 1.8 | 97.50 |
| Accuracy (%) | XGBoost | **94.73** | 2.5 | **98.18** |
| | Random Forest | **94.73** | 2.5 | **98.18** |
| | Decision Tree | 63.01 | 5.7 | 80.36 |
| | Support Vector | 92.16 | 2.5 | 98.00 |
| | Naïve Bayes | 93.46 | 2.0 | 98.00 |
| | Logistic Regression | 93.63 | 1.9 | 96.36 |
| AUC Score (%) | XGBoost | **98.78** | 0.8 | **100.00** |
| | Random Forest | 84.99 | 4.9 | 93.59 |
| | Decision Tree | 59.95 | 6.0 | 77.96 |
| | Naïve Bayes | 97.92 | 1.4 | 99.67 |
| | Logistic Regression | 98.68 | 0.7 | 99.67 |

extensive CV analysis not only provides a more robust assessment of model performance but also offers insights into the variability of results. The comprehensive findings affirm that XGBoost consistently outperformed other models, showcasing both the best and most reliable results among the experimented models.

A calibration curve shown in Fig. 5 is a plot of predicted probabilities against the actual fraction of positive classes in a dataset. It is a useful tool for evaluating the calibration of a classification model, which refers to how well the predicted probabilities reflect the actual probabilities of a positive outcome and may help us interpret how decisive a classification model is. The calibration curves for the XGBoost model as well as baseline methods, including logistic regression, random forest, SVC, decision tree, and naïve Bayes. This comprehensive representation allows for a detailed comparison of calibration performance across various models. The calibration curves collectively showcase the superior performance of the XGBoost model in achieving optimal accuracy for the classification of ME/CFS patients.

In this study, the receiver operating characteristic (ROC) curve was employed as a pivotal metric to evaluate and compare the performance of the proposed model against various alternative algorithms. The ROC curve serves as a graphical representation, illustrating the efficacy of binary classification models by plotting the recall, or true positive rate (TPR), against the false positive rate (FPR) across different threshold settings. This curve is particularly instrumental in assessing model performance in scenarios where class distribution is imbalanced. An optimal model is characterized by an ROC curve that approximates the upper left corner of the plot, indicative of a high TPR and a low FPR. The empirical results of this analysis are depicted in Fig. 6, which includes the area under

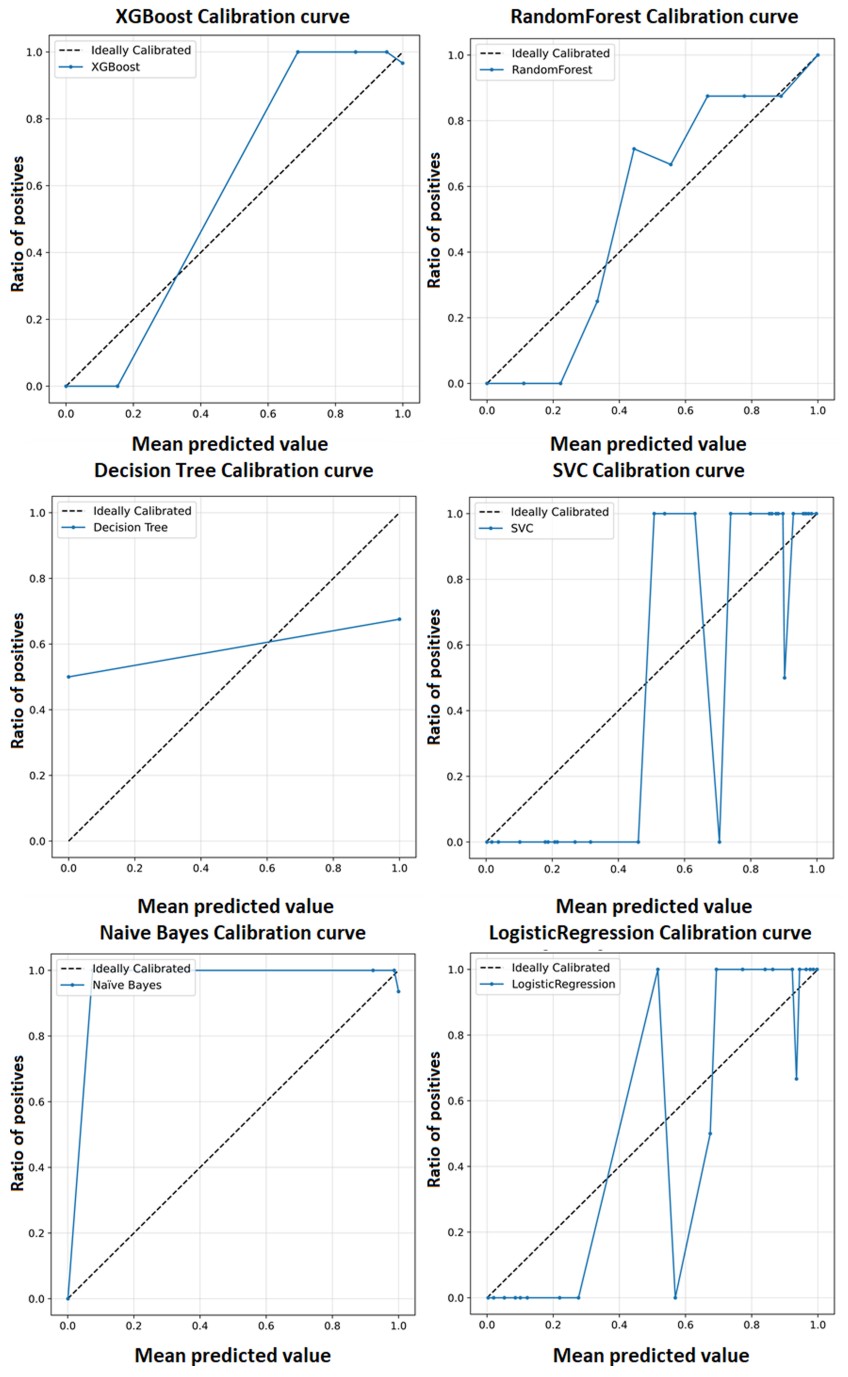

**Figure 5** Calibration curve for the proposed model (XGBoost).

the ROC curve (AUC-ROC) for each model. The AUC metric, a widely acknowledged standard for model evaluation, quantifies the two-dimensional area underneath the entire ROC curve. Higher AUC values are indicative of superior model performance, reflecting a model's ability to distinguish between the classes with greater accuracy. The XGBoost

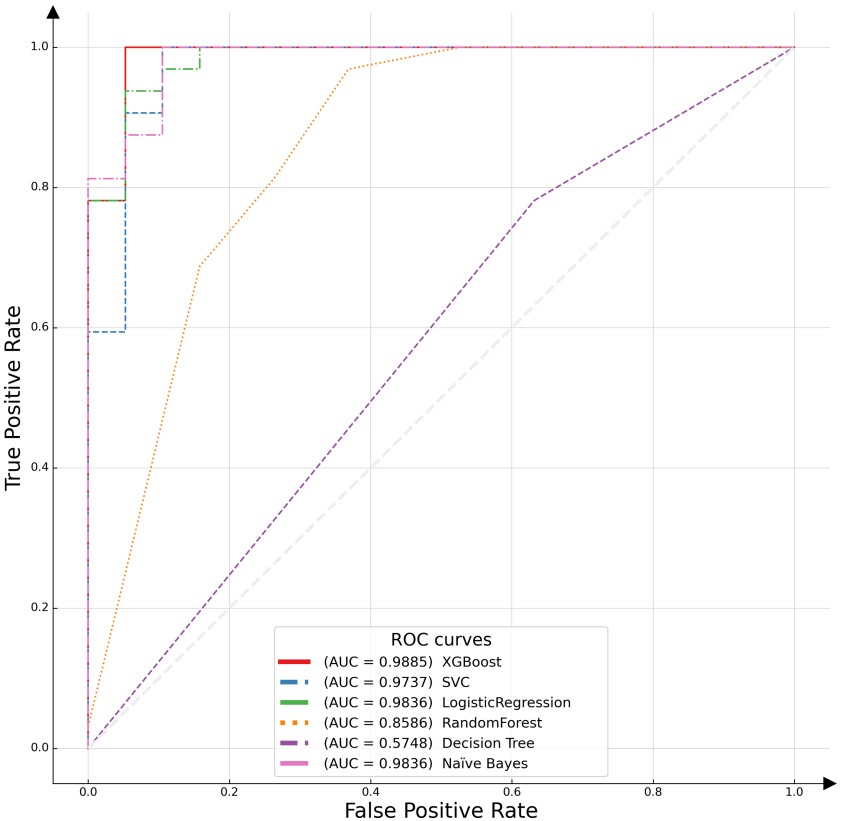

**Figure 6    ROC curves of different classifiers.**

algorithm emerged as the most effective model, demonstrating the highest AUC value, followed by logistic regression. This distinction underscores the relative predictive strengths of these models within the context of our binary classification task.

Within the scope of this study, the SHAP was applied to the model that was generated by XGBoost. SHAP plots are an effective method for comprehending the role that each individual feature plays in the process of predicting the output variable. Both the SHAP summary plot (Fig. 7) and the SHAP beeswarm plot (Fig. 8) showed that 'alpha-CEHC sulfate' was the metabolite in the dataset that had the greatest significance for ME/CFS prediction. Moreover, decreased levels of alpha-CEHC sulfate, hypoxanthine, and phenylacetylglutamine, and increased levels of N-delta-acetylornithine, and oleoyl-linolooyl-glycerol (18:1/18:2) [2] were found to increase the risk of ME/CFS. This information can be put to use in the process of gaining a deeper comprehension of the underlying mechanisms that are responsible for ME/CFS patients.

## DISCUSSION

The current article presents an ML framework to extract potential metabolic biomarkers and predict ME/CFS based on these biomarkers in ME/CFS patients. In this ML framework, combinations of feature selection, ML models, and XAI were explored in depth to create

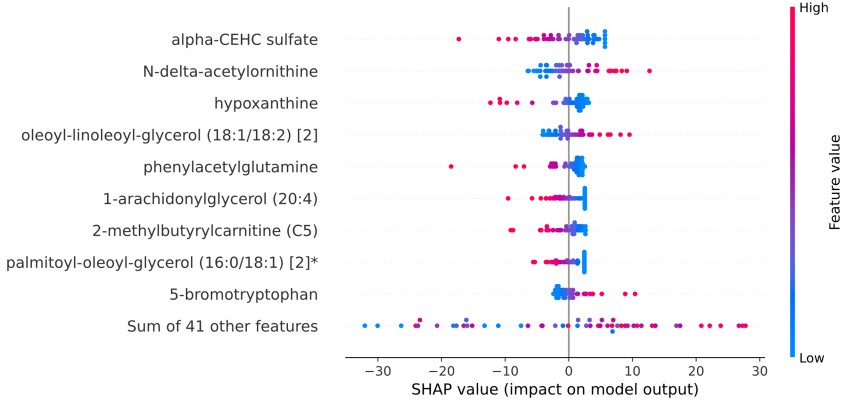

**Figure 7    SHAP summary plot.**

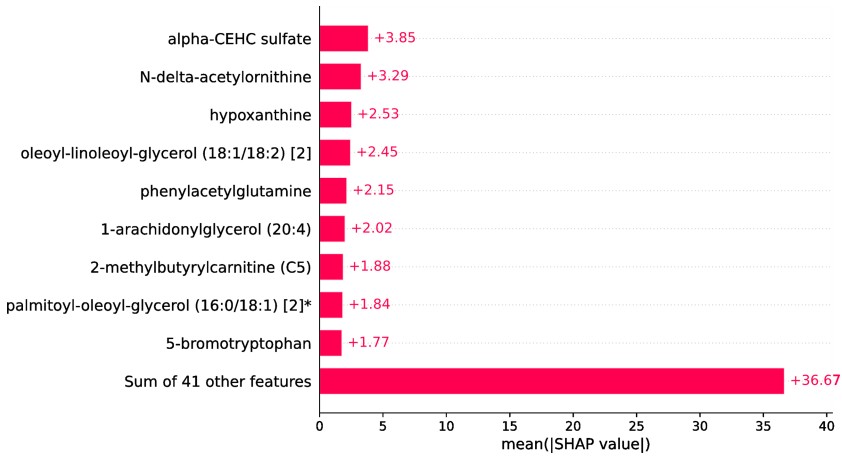

**Figure 8    SHAP beeswarm plot.**

a metabolomics-based diagnostic model in ME/CFS patients. XGBoost, SVM, logistic regression, random forest, decision tree, and Naïve Bayes classifiers were created based on all standardized data features and biomarker candidate metabolite features determined after feature selection. When all features (832 metabolomic features) were employed, logistic regression performed best with 80.36% accuracy and a Brier score of 0.143. However, to increase the prediction performance and to obtain a more interpretable model, feature selection was applied to reduce the number of biomarkers, and the results were compared. Performance improved after ANOVA-based feature selection, and of the six ML classifiers, XGBoost showed the best performance and calibration (98.18% accuracy, 98.85% AUROC, 0.023 brier score, and 98% F1-score) with fast computation and strong generalization ability. It was observed that the accuracy of the model increased by 18% after feature selection, and therefore, the XGBoost model was used as the final model for ME/CFS estimation. Furthermore, the XGBoost model obtained 100% sensitivity. A greater sensitivity value indicates a lower false negative (FN) score. False positive and false

negative mistakes are widespread in comparative biological studies. As a result, it is critical to establish the likelihood that an actual effect is material (*Li et al., 2020*). A lower FN score is a good sign for ME/CFS patients. This finding is critical since one of the primary aims of this study is to reduce missing instances of ME/CFS (false negatives). Furthermore, we extended the evaluation by performing CV 50 times to increase the robustness and reliability of model comparisons. When the average, standard deviation, and best results obtained for each model over 50 CV periods are examined, it is confirmed that XGBoost consistently outperforms other models and achieves both the best and most reliable results among the tested models. The XAI approach, SHAP, was used to explain the model's decisions in the prediction of ME/CFS. The SHAP results indicated that decreased levels of alpha-CEHC sulfate, hypoxanthine, and phenylacetylglutamine, as well as increased levels of N-delta-acetylornithine and oleoyl-linoloyl-glycerol (18:1/18:2)[2] increased the risk of ME/CFS.

*Hoel et al. (2021)* analyzed ME/CFS metabolic phenotypes using global metabolomics, lipidomics, and hormone measurements. The researchers analyzed serum samples from 83 ME/CFS patients and 35 healthy controls. They found that certain metabolic changes were common among the patients, indicating increased energy strain and altered utilization of fatty acids and amino acids as energy sources. The metabolites were clustered into three groups (metabolite blocks) with different discriminating impacts on the metabotype (*Hoel et al., 2021*). In this study, ME/CFS samples were classified using the identified metabolic biomarkers, and these metabolites varied between patients and controls.

According to the results, one of the biomarker candidate metabolites for ME/CFS patients was 1-arachidonylglycerol (20:4), which is an endocannabinoid, a type of molecule involved in regulating cognitive and physiological processes in the body, including energy balance, emotion, pain sensation, and neuroinflammation (*Kohansal et al., 2022*). The other important metabolite was Alpha-hydroxyisovalerate (AHIV). *Mukherjee et al. (2017)* reported AHIV, among several other metabolites, higher in head and neck squamous cell carcinoma (HNSCC) compared to normal cases. Samples were obtained from cancer patients and healthy participants (*Mukherjee et al., 2017*) as mouthwash. Hypoxanthine was another important metabolite in ME/CFS. Some studies have reported increased uric acid levels in individuals with ME/CFS compared to healthy control samples. These findings point to potential disturbances in purine metabolism, including hypoxanthine (*Armstrong et al., 2015*; *Zolkipli-Cunningham et al., 2021*). The findings of the study herein complement the reported studies, particularly in identifying Alpha-CEHC sulfate and N-delta-acetylornithine as key metabolites in ME/CFS. Alpha-CEHC sulfate, a vitamin E metabolite, and N-delta-acetylornithine, involved in the urea cycle and amino acid metabolism, are significant for their roles in oxidative stress regulation and metabolic pathways, respectively. These studies collectively emphasize the role of metabolic disturbances in ME/CFS, pointing to a multifaceted understanding of the disease that could guide future diagnostic and therapeutic strategies.

According to the SHAP results from the current work, it was determined that the model was important in the decision-making process for the classification of ME/CFS of alpha-CEHC sulfate and N-delta-acetylornithine metabolites. Alpha-CEHC sulfate, also known as

alpha-tocopherol quinone, is a metabolite of vitamin E. It has antioxidant properties and is involved in oxidative stress regulation (*Traber & Atkinson, 2007*). N-delta-acetylornithine, also known as N-acetylornithine or NAO, is a metabolite involved in the urea cycle and amino acid metabolism (*Liu et al., 2019*).

To address benchmarking against prior results, the current study indeed provides a comprehensive comparison of the proposed method with other classifiers and presents a detailed comparative analysis (as described in Table 3), conducted through 50 times CVs, to demonstrate the robustness and reliability of the model comparisons. This extensive analysis not only assesses the performance of the proposed XGBoost model; but also compares it with other models such as RF, DT, SVM, GNB, and LR. The findings, particularly focusing on metrics like the F1-Score, show that XGBoost consistently outperformed these models, providing both superior and more reliable results.

The conducted study introduces a novel methodological framework that amalgamates ML with XAI. The merit of this approach primarily resides in its capacity to enhance the interpretability of complex ML models, particularly through the incorporation of SHAP. This integration is innovative as it not only facilitates the identification of key metabolic biomarkers associated with ME/CFS; but also elucidates their respective contributions to the predictive model. Further, the adoption of the XGBoost classifier, recognized for its efficiency in handling structured data, underscores the robustness of the proposed methodology. The study's novel use of feature selection techniques significantly streamlines the model by reducing the number of necessary biomarkers, thereby enhancing both the model's practicality for clinical applications and its interpretability. The methodological rigor is exemplified by the model achieving an exceptional accuracy rate (98.18%) and a high AUCROC value (98.85%). This combination of high predictive accuracy with advanced interpretability, through the use of XAI, marks a significant contribution to the field, potentially paving the way for more accurate and comprehensible diagnostic tools for ME/CFS. The achieved high accuracy and AUCROC values substantiate the efficacy and novelty of our methodology, reflecting a substantial contribution to the field of medical diagnostics.

The study had some limitations. The first limitation is that the data set used is large in terms of feature sizes but small in terms of sample size. Especially in the era of big data healthcare, where there are extremely high dimensional features and large sample sizes, ML and XAI can better reflect powerful learning capabilities. Therefore, more research is needed and more data needs to be collected for a deeper study of ME/CFS. The second limitation was not having external validation data, but an attempt was made to check for problems caused by this limitation using cross-validation. In the study, patients' metabolomic data were analyzed and ME/CFS was estimated based on these data. More detailed research is needed to integrate clinical risk factors, environmental factors, lifestyles, and other factors related to ME/CFS to improve future predictions and examine the impact of confounding factors. Further, while the literature indicates that biomarker and predictive research articles in ME/CFS are limited, further research and wet lab experiments are required to better understand the relationships between findings and disease. The public dataset in the study provides information on age and BMI, but omits other crucial demographic details

like stress levels and working hours. Future research incorporating a broader spectrum of demographic and lifestyle variables may enhance the model's ability to distinguish ME/CFS-specific metabolite molecules as indicative of ME/CFS.

# CONCLUSIONS

ME/CFS biomarkers are one of the most urgently needed developments in this field as a means of definitive diagnosis and monitoring the effectiveness of treatments. With this need in mind, the current study focused on the interpretable classification of XAI-based ME/CFS. The results provided a first step toward establishing prognostic models for the classification of ME/CFS. Future studies in which a larger and independent cohort is analyzed and compared with other exhausting diseases can likely increase the confidence of classification and reveal whether plasma metabolomics can serve as a reliable tool for objective identification and monitoring of ME/CFS patients.

## Funding

This research project is funded by Princess Nourah bint Abdulrahman University Researchers Supporting Project number (PNURSP2024R309), Princess Nourah bint Abdulrahman University, Riyadh, Saudi Arabia. The funders had no role in study design, data collection and analysis, decision to publish, or preparation of the manuscript.

## Grant Disclosures

The following grant information was disclosed by the authors:
Princess Nourah bint Abdulrahman University, Riyadh, Saudi Arabia: PNURSP2024R309.

## Competing Interests

The authors declare there are no competing interests.

## Author Contributions

- Fatma Hilal Yagin conceived and designed the experiments, performed the experiments, analyzed the data, performed the computation work, prepared figures and/or tables, authored or reviewed drafts of the article, and approved the final draft.
- Ahmadreza Shateri conceived and designed the experiments, performed the experiments, analyzed the data, performed the computation work, prepared figures and/or tables, authored or reviewed drafts of the article, and approved the final draft.
- Hamid Nasiri conceived and designed the experiments, performed the experiments, analyzed the data, performed the computation work, prepared figures and/or tables, authored or reviewed drafts of the article, and approved the final draft.
- Burak Yagin conceived and designed the experiments, performed the experiments, analyzed the data, performed the computation work, authored or reviewed drafts of the article, and approved the final draft.

- Cemil Colak conceived and designed the experiments, performed the computation work, authored or reviewed drafts of the article, and approved the final draft.
- Abdullah F Alghannam conceived and designed the experiments, authored or reviewed drafts of the article, and approved the final draft.

## Ethics

The following information was supplied relating to ethical approvals (i.e., approving body and any reference numbers):

Inonu University Non-Interventional Clinical Research

## Data Availability

The raw data and codes are available in the Supplementary Files.

## Supplemental Information

Supplemental information for this article can be found online at http://dx.doi.org/10.7717/peerj-cs.1857#supplemental-information.

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
