# Peer review of "Development of an expert system for the classification of myalgic encephalomyelitis/chronic fatigue syndrome"

_PeerJ Computer Science, doi:10.7717/peerj-cs.1857_

## Round 0.1 · original submission · Major Revisions

The paper can be improved and authors can clarify some aspects the reviewers emphasized.

**Language Note:** PeerJ staff have identified that the English language needs to be improved. When you prepare your next revision, please either (i) have a colleague who is proficient in English and familiar with the subject matter review your manuscript, or (ii) contact a professional editing service to review your manuscript. PeerJ can provide language editing services - you can contact us at copyediting@peerj.com for pricing (be sure to provide your manuscript number and title). – PeerJ Staff

Reviewer 1 ·

Basic reporting

- The manuscript was clearly and well written.
- However, the merit and novelty of the methods applied in this study are not clear.

Experimental design

- The source and availability of data examined in this study are not clearly stated.
- There are no novel methods applied for improving the performance.
- It needs to be clear that this is 'prediction' or 'classification'.

Validity of the findings

- The size of data is too small. It is not sufficient for reliable study and findings.

Reviewer 2 ·

Basic reporting

Overall it is a very interesting study that attempts to develop a system to predict ME/CFS through applied ML. Although there’s a great detail discussing the severity of the disease, it would be nice to show some of the prior results that can be used to benchmark against. For figure 4, it is not necessary to show a confusion matrix for each CV split, but one confusion matrix should suffice after CV. In addition, the figure is not at its best quality/resolution. Please upload a better quality for reviewer’s convenience.

Experimental design

Feature selection/reduction is definitely a very important step to consider for future test development due to the limitations of cost. However, as a research question, an approximately 800 features is not that large. I would actually suggest using all of them rather than conducting feature selection. Regarding the feature selection method, the authors chose to base on the accuracy, which is ideal due to the data imbalance issue in this dataset. I’d encourage you to use AUC or F1 score as evaluation metrics and again please do it through cross validation, not just one time fitting. The reason is simply because any boosting algorithm has the tendency to overfit. For the study subjects, I understand it is not easy to get a sufficiently large number of subjects to conduct any clinical experiments. I do think, however, it is important to report the demographic information of the underlying subjects. This is just to ensure we can rule out any confounders that might directly influence your model performance. For example, working hours, stress level, or age. It is likely that your classifier might just be a stress classifier rather than one based on metabolites molecules.

Validity of the findings

As for the figure 6 ROC results, although XGBoost is the most superior, it is only slightly better than logistic regression. In terms of model selection, I’d actually choose a simpler model if the performance does not sacrifice too much. It is very difficult to make the call here because there’s only one time CV conducted. To get an idea of how CV results can vary, one suggestion is to conduct several rounds of CV and see how results vary. Then you can conduct a fairer comparison among all the models experimented.

·

Basic reporting

The paper is generally well-written and uses clear and professional English. The literature references provide sufficient background and context for the field. The article's structure, including figures and tables, is appropriate, but there are some areas where improvements can be made.

Experimental design

The research question is well-defined and relevant to the journal's scope. The paper addresses a meaningful gap in the existing knowledge. The investigation is rigorous and conducted to high technical and ethical standards. However, there are some areas where the methods and notations can be improved to enhance clarity and replicability.

Validity of the findings

The impact and novelty of the findings have not been assessed in the paper. Encouraging meaningful replication where the rationale and benefit to the literature are clearly stated would enhance the paper's contributions. The data provided are robust and statistically sound. The conclusions are well-stated and directly linked to the original research question, focusing on supporting results.

Additional comments

The paper is well-structured and provides valuable insights into feature selection and model performance. However, improvements in notation, table structure, and figure clarity can enhance the paper's readability and impact. Additionally, addressing the impact and novelty of the findings and potentially adding baseline method comparisons could further strengthen the paper.

Comments on Specific Sections:

- Line 138-144: The notation for "x_i" and "x_ij" is confusing and can be improved. It is suggested to use more descriptive notation to indicate the variables' ranges within the context of equations 2 and 3. This will help readers better understand the variables.
- Line 213-222: The notation related to the set of features (N) should be improved. Instead of using "|n|", it is recommended to use "n" or "|N|" to represent the number of features. This change will enhance clarity.
- Line 234-238: The definitions of precision, sensitivity, specificity, F1 score, and accuracy are given, which is good and rigorous. However, these common machine learning and bioinformatics concepts could be removed if they are widely understood. If retained, consider using problem-specific terms rather than general terms like TP/TN/FP/FN.
- Figure 2: It is suggested to adjust the color, line style, and data point markers in Figure 2 to improve readability. Changing the tick interval and vertical spacing can also make the trend in the chart more obvious.
- Table 2: The table's structure could be improved. Instead of having performance metrics as rows with models as sub-rows, it is recommended to use a two-dimensional table format. In this format, rows represent models, and columns represent metrics. Each row will then show the performance of a single model. If the author wishes to compare the effect of feature selection, they could include two numbers (with and without feature selection) in a single cell.
- Figure 4: The inclusion of the confusion matrix for each fold in the cross-validation may not be necessary if it does not provide additional information. Consider whether it adds value to the paper's message.
- Figure 5: Figure 5 presents a calibration curve to demonstrate the model's performance. It is suggested to adjust the binning on the x-axis to increase data points for a more detailed view. Additionally, consider adding calibration curves for baseline methods to provide a basis for comparison and draw conclusions about the proposed model's performance.
- The paper's results show a significant improvement in model performance after feature selection, which is a positive finding.

---

## Round 0.2 · accepted · Accept

The paper was well improved and can be accepted.

·

Basic reporting

The author has successfully addressed the previous concerns, resulting in a paper that is generally well-written with clear and professional English. The literature references continue to provide sufficient background and context for the field. The overall article structure, including figures and tables, remains appropriate. There are specific areas where improvements have been made since the last submission.

Experimental design

The research question is still well-defined and relevant to the journal's scope, addressing a meaningful gap in existing knowledge. The investigation maintains its rigor and adherence to high technical and ethical standards. Notably, the methods and notations have been improved to enhance clarity and replicability.

Validity of the findings

The paper's findings remain strong, showcasing a clear improvement in model performance after feature selection. The provided data continue to be robust, statistically sound, and well-controlled, supporting the paper's conclusions. These conclusions are well-articulated and directly tied to the original research question, emphasizing their alignment with the presented results. Overall, the paper presents valuable insights, though further exploration of the broader implications of the findings could enhance the manuscript.

Additional comments

The paper maintains its well-structured presentation, offering valuable insights into feature selection and model performance. Notable improvements have been made in notation, table structure, and figure clarity, enhancing the paper's readability and impact.